# *NB-LRRs* Not Responding Consecutively to *Fusarium oxysporum* Proliferation Caused Replant Disease Formation of *Rehmannia glutinosa*

**DOI:** 10.3390/ijms20133203

**Published:** 2019-06-29

**Authors:** Aiguo Chen, Li Gu, Na Xu, Fajie Feng, Dexin Chen, Chuyun Yang, Bao Zhang, Mingjie Li, Zhongyi Zhang

**Affiliations:** 1Key Laboratory of Ministry of Education for Genetics, Breeding and Multiple Utilization of Crops, Fujian Agriculture and Forestry University, Fuzhou 350002, China; 2Institute of Tobacco Research, Chinese Academy of Agricultural Sciences, Qingdao 266101, China

**Keywords:** *Rehmannia glutinosa* L., replant disease, rhizosphere microbes, *NB-LRR*, plant hormone

## Abstract

Consecutive monoculture practice facilitates enrichment of rhizosphere pathogenic microorganisms and eventually leads to the emergence of replant disease. However, little is known about the interaction relationship among pathogens enriched in rhizosphere soils, Nucleotide binding-leucine-rich repeats (NB-LRR) receptors that specifically recognize pathogens in effector-triggered immunity (ETI) and physiological indicators under replant disease stress in *Rehmannia glutinosa*. In this study, a controlled experiment was performed using different kinds of soils from sites never planted *R. glutinosa* (NP), replanted *R. glutinosa* (TP) and mixed by different ration of TP soils (1/3TP and 2/3TP), respectively. As a result, different levels of TP significantly promoted the proliferation of *Fusarium oxysporum* f.sp. *R.*
*glutinosa* (FO). Simultaneously, a comparison between FO numbers and *NB-LRR* expressions indicated that *NB-LRRs* were not consecutively responsive to the FO proliferation at transcriptional levels. Further analysis found that *NB-LRRs* responded to FO invasion with a typical phenomenon of “promotion in low concentration and suppression in high concentration”, and 6 *NB-LRRs* were identified as candidates for responding *R. glutinosa* replant disease. Furthermore, four critical hormones of salicylic acid (SA), jasmonic acid (JA), ethylene (ET) and abscisic acid (ABA) had higher levels in 1/3TP, 2/3TP and TP than those in NP. Additionally, increasing extents of SA contents have significantly negative trends with FO changes, which implied that SA might be inhibited by FO in replanted *R. glutinosa*. Concomitantly, the physiological indexes reacted alters of cellular process regulated by NB-LRR were affected by complex replant disease stresses and exhibited strong fluctuations, leading to the death of *R. glutinosa*. These findings provide important insights and clues into further revealing the mechanism of *R. glutinosa* replant disease.

## 1. Introduction

Replant diseases, also known as consecutive monoculture problems or sick soil syndrome, are widespread in the production of different crops, especially in medicinal crops, fruit tree and vegetables, such as *Rehmannia glutinosa*, *Panax notoginseng*, apples, peach, strawberry, soybean etc. [1,2,3,4,5,6], which usually lead to disease aggravation, poor growth status, yield reduction and quality deterioration [7]. *Rehmannia glutinosa* Libosch, a perennial herbaceous plant, is one of the 50 traditional Chinese medicines with high value [8]. However, consecutive monoculture of *R. glutinosa* results in its abnormal growth and a significant decline in the yield and quality of tuberous roots. Moreover, it cannot be replanted on the same land for 8–10 years [9]. The causes of problems associated with consecutive monoculture of *R. glutinosa* have become a research priority in China [10].

Allelochemicals, soil-borne diseases and soil quality deterioration are usually considered as primary factors involved in the formation of replant disease [11]. Some studies concluded that the effect of soil chemical properties is inconsistent with replant disease over time [12,13]. Furthermore, allelochemicals, such as phenolic acid and flavonoids, are quickly metabolized by soil microorganisms and shape the composition and diversity of microbial community in rhizosphere soils [14,15,16,17,18,19]. Recent studies have appointed that the healthy growth of plants is closely related to the balance of the rhizosphere microbes [20,21,22]. There is mounting evidence that the biotic factors mediated by rhizosphere allelochemicals is a causal agent of replant disease [10,23,24,25]. Previous studies have discovered that the pathogenic microorganisms in replanted *R. glutinosa* rhizosphere soil were selectively attracted by root exudates, and then colonized the root surface to proliferate, resulting in rhizosphere micro-ecology catastrophe, mainly in the transformation of rhizosphere microorganisms from “bacterial” to “fungal” types and a decline in rhizosphere microbial diversity [10,11,13,26,27]. The previous evidences have confirmed that the pathogenic *Fusarium oxysporum* f.sp. *R. glutinosa* (FO) was enriched and beneficial *Pseudomonas* spp. (PS) specifically decreased in rhizosphere soil of replanted *R. glutinosa* [28,29,30]. However, it remains largely unknown the mechanism of how replanted *R. glutinosa* responded to the changes in composition and diversity of microbial community in rhizosphere soil.

In natural environments, plants can regulate the rhizosphere micro-ecological status to preserve healthy growth by constantly renewing the composition and diversity of microbiome [31]. Plant immune response, containing recognition to pathogens and signal transduction, thus plays a key role in coordinating microbial communities in rhizospheres [32,33,34]. As we know, plant innate immunity co-evolving with pathogenic microbes have developed two strategies, including pathogen-associated molecular patterns (PAMP)-triggered immunity (PTI) and effector-triggered immunity (ETI) [35,36]. PTI is generally effective against non-adapted pathogens in a phenomenon called non-host resistance, and the pattern recognition receptors (PRRs) of PTI systems can recognize conserved PAMP features of different species or genera. ETI is active against adapted pathogens, and the receptors of ETI systems specifically and robustly respond to pathogen effectors through nucleotide-binding-leucine-rich repeat (NB-LRR) domain-mediated perception [37]. NB-LRRs are now believed to include the majority of plant R proteins and recognize fast-evolving effectors [38,39]. Our previous studies found that pathogenic FO was specifically enriched in the rhizosphere soil of replanted *R. glutinosa* [28], and PTI of two plant innate immunity silently responded to replant disease [40], which suggested that PTI was not sufficient to cope with fast-evolving effectors. However, some NB-LRR receptors in ETI were upregulated [41]. It remains unclear why these upregulated NB-LRRs fail to prevent the death of replant *R. glutinosa*.

Plant hormones are essential regulators for triggering plant immune resistance to cope with various pathogens [42]. The salicylic acid (SA) and jasmonic acid-ethylene (JA-ET) are believed to form the hormonal backbone of plant immune responses to pathogens, with SA involved in resistance to biotrophic pathogens and JA–ET involved in responses to necrotrophic pathogens and chewing insects [37,43]. The antagonism between SA and JA-ET often occurs through the regulatory protein non-expressor of pathogenesis-related proteins 1 (NPR1), which mediates the SA-induced expression of pathogenesis-related (PR) genes and systemic acquired resistance (SAR) [44]. Enhanced disease susceptibility 1 (EDS1) acts as a SA-pathway inducer and a JA-ET pathways repressor, and mitogen-activated protein kinase 4 (MPK4) acts as a negative regulator of systemic acquired resistance, which is controlled by the SA pathway [37]. In addition, abscisic acid (ABA) can increase susceptibility to pathogens to some extent, although it is mainly associated with responses to abiotic stresses [37,45]. Some studies have described that SA was involved in the defense against *Fusarium oxysporum* and associated with acteoside accumulation, one of important pharmacodynamic component [46,47]. Our recent study displayed that plant hormones involved in the formation mechanism of *R. glutinosa* replant disease and are closely related to immune resistance [48]. However, a comprehensive survey on the relationship between plant hormones and replant disease in *R. glutinosa* is still unknown. 

Given the interferences from complex field environment and vegetative propagation, such as various pathogens, variable light and temperature conditions, maternal resistance, size of vegetative mass, germination rate and site of shoot, pot experiment in controlled conditions with regenerated plantlets is thus an ideal research method. Here, the soils from the sites in which *R. glutinosa* had never been planted and had been consecutively planted *R. glutinosa* were proportionally mixed to control the stress levels of replant disease. The regenerated plantlets of *R. glutinosa* with feeble immunity were acclimatized and transplanted in a phytotron. The dynamic changes of pathogenic microbes in the rhizosphere soils were investigated. At same time, *NB-LRRs* expression, plant hormone contents and physiological index levels were examined in the roots of *R. glutinosa*. These research results will provide important insights for further revealing the mechanism of formation of replant disease.

## 2. Results

### 2.1. Changes in the Numbers of Pseudomonas spp. and Fusarium oxysporum in Rhizosphere Soils of Replant Disease R. glutinosa

To understand the changes in the beneficial PS and pathogenic FO in rhizospheres of replanted *R. glutinosa*, the numbers of them were showed in Figure 1. The numbers of PS were 3.87 × 10^8^~5.99 × 10^8^ (cell·g^−1^ soil), the numbers of FO were 0.71 × 10^8^~2.68 × 10^8^ (cell·g^−1^ soil), and the ratio of FO to PS were 0.16~0.55. The maximum difference ranged between 0 DAP and 9 DAP, where PS was in TP soil and FO was in NP soil (Figure 1A,B). Moreover, only the numbers of FO decreased significantly over time in control NP (Figure 1E). The inhibiting effect of NP soil on FO proliferation was thus stronger than the promoting effect of replanted soil on FO proliferation. During 0~9 DAP, the FO numbers and ratio of FO to PS in TP soil were significantly lower than that in NP soil at 0 DAP, and the reversing results with significant difference were at 9 DAP (Figure 1B,C). The PS numbers were significantly lower only in NP soil than that in TP soil at 0 DAP, and the reversed trends with no significant difference were at 9 DAP (Figure 1A). Furthermore, there were the significant positive correlations between the addition level of replant soils and the changes of the FO numbers and the FO/PS values, and between the FO numbers and the FO/PS values (Table 1 and Table 2). The results evidenced that *R. glutinosa* replant disease (1/3TP, 2/3TP and TP) could significantly promote the FO proliferation in rhizospheres soils. Interestingly, at 0 DAP, the PS numbers in 1/3TP, 2/3TP and TP soil were 1.04, 1.11 and 1.21 times than that in NP soil respectively, and the FO numbers in NP soil were 1.39, 1.44 and 1.83 times than that in 1/3TP, 2/3TP and TP soil respectively. The changes were obviously different from that at 9 DAP. The results suggested that the soil had a strong ability that could restore balances among microbial communities, and this means that the appropriate ration between FO and PS in rhizosphere soil presented in *R. glutinosa* grows healthly, during fallow stage.

### 2.2. NB-LRR Lists Response to Replant Disease Stresses in R. glutinosa

To reveal the responding mode of *R. glutinosa NB-LRRs* during replant disease formation, the expression levels of *NB-LRRs* were determined by qRT-PCR. Of 35 expressed *NB-LRRs* in acclimatization stage, seven were upregulated and 11 were downregulated (Figure 2). In contrast to NP, 22 *NB-LRRs* were upregulated and six *NB-LRRs* were downregulated in TP at 3 DAP, which accounted for 80% of the 35 *NB-LRRs*. The stage from 0 to 3 DAP was thus the key stage to *NB-LRRs* that responded to replant disease stress. In addition, only 1 *NB-LRR* was upregulated and 29 *NB-LRRs* were downregulated at 6 DAP, indicating that replant disease interference to the expression of *NB-LRRs*. In 3 DAP, the Pearson correlation analyses displayed that 35 *NB-LRRs* had no significant correlation with the PS numbers. The significant positive correlation was only between 12 *NB-LRRs* and the FO numbers in NP, 1/3TP, 2/3TP (Table 3). However, these *NB-LRRs* that have significantly positive correlation with the FO numbers, have ineffectively prevented the death of *R. glutinosa* plants at 6 DAP. The results displayed that the 12 upregulated *NB-LRRs* might not respond to the pathogenic FO in replanted *R. glutinosa*. Therefore, the six downregulated *NB-LRRs* (*RgNB5*, *RgNB14*, *RgNB26*, *RgNB29*, *RgNB34* and *RgNB35*) in TP at 3 DAP, were screened as candidates for responding to *R. glutinosa* replant disease. In addition, *RgNB14* and *RgNB26* of the 6 *NB-LRRs* were upregulated expression, and the remaining 4 *NB-LRRs* were downregulated or presented a decreasing trend at expression level in acclimatization stage. The two groups of *NB-LRRs* represented whether they responded to abiotic stress or not. Noticeably, the expression levels of *RgNB5* and *RgNB29* were continuously downregulated with addition levels of replant soils at 3 DAP.

### 2.3. Plant Hormones Response to Replant Disease Stresses in R. glutinosa

Four plant hormones including ABA, ET, JA and SA were measured, and their contents in 1/3 TP, 2/3TP and TP treatments fluctuated obviously from 0 DAA to 9 DAP (Figure 3). In the acclimatization stage (from 0 DAA to 3 DAA), the contents of plant hormones decreased significantly except for SA. The sensitive stages for rapid alteration of hormones was from 0 to 3 DAP based on the comparison between 3 DAP and 3 DAA (0 DAP). At 3 DAP, the four hormones were significantly activated under replant disease stresses compared with the control NP by 0.56 to 1.53 times for ABA, 0.59 to 2.02 times for JA, 0.50 to 1.50 times for ET and 0.37 to 1.19 times for SA (*p* < 0.05), which showed that the four plant hormones responded strongly to replant disease stress levels. From 0 to 3 DAP, the four hormone contents in 1/3TP, 2/3TP and TP have significantly increased at 3 DAP compared with that at 0 DAP, excepting SA treated with TP soils. Of which, ABA contents increased by 65.26% to 173.64%, JA by 60.19% to 203.50%, ET by 65.26% to 173.84% and SA by 3.91% to 66.39% under replant disease stresses, but only increased by 8.50%, 0.33%, 9.73% and −24.72% in NP, respectively (*p* < 0.05). Notably, there were significant negative correlations between FO and SA (*p* < 0.05) and significant positive correlations among the four hormones (*p* < 0.01) (Table 4). The results showed that the FO in replant disease rhizosphere soil inhibited the SA biosynthesis in root. Taking together these changes showed that the complex replant disease stress, including biotic and abiotic stress, promoted the four hormones release, but FO inhibited SA biosynthesis, resulting in *R. glutinosa* death at 6 DAP.

### 2.4. Physiological Response to Replant Disease Stresses in R. glutinosa

The physiological indexes including root activity; superoxide dismutase (SOD), peroxidase (POD) and catalase (CAT) activities; and malondialdehyde (MDA) and hydrogen peroxide (H_2_O_2_) contents were investigated in *R. glutinosa* with different replant disease stress levels (Figure 4). In the acclimatization stage, there were no significant changes in the POD activities. The H_2_O_2_ contents were significantly increased, and the others that used to eliminate toxicity of H_2_O_2_ were decreased significantly. In contrast to 3 DAA (0 DAP), the root activities, SOD activities, POD activities and MDA contents in NP tended to increase consecutively from 3 DAP to 9 DAP, which reported the normal response of plantlets planted in healthy soil. In addition, root activity, SOD activity, CAT activity and H_2_O_2_ contents in NP, 1/3TP, 2/3TP and TP at 3 DAP were significantly different from those at 3 DAA (0 DAP). Of which H_2_O_2_ contents significantly decreased by 0.45 to 0.75 times, and root activity, SOD activity and CAT activity significantly increased by 0.36 to 2.75 times, 3.12 to 7.33 times, 2.76 to 4.26 times at 3 DAP, respectively. However, there were no gradient changes for the four physiological indexes at 3 DAP. Moreover, only the H_2_O_2_ contents showed a gradient change at 6 DAP. Furthermore, there were significant positive correlations between the FO numbers and the CAT activity and the MDA content only at 6 DAP (Table 5). Overall, these results displayed that these physiological indexes acting downstream of life activities were affected by complex replant disease stresses and did not show significant correlation with the FO numbers until 6 DAP. 

## 3. Discussion

### 3.1. Replant Disease Promotes Proliferation of Fusarium oxysporum in R. glutinosa Rhizospheres Soil

Previous studies have revealed that *R. glutinosa* replant disease induced rhizosphere microbes’ adverse chemotaxis [49,50], resulting in the increase of pathogenic FO abundance and the decrease of beneficial PS abundance, thus the identified PS and FO were usually used as a characteristic mark in the *R. glutinosa* replant disease study [10,28]. In this study, the PS numbers decreased and the FO numbers increased in 1/3TP, 2/3TP and TP soils from 0 to 9 DAP. Moreover, the significant positive correlations were presented between the addition levels of replant soils and the changes of the FO numbers and the FO/PS values. These results revealed that replant disease promoted FO proliferation in rhizospheres soils. Interestingly, the focus on maximum difference between the PS and FO numbers displayed that the inhibiting effect of NP soil on FO proliferation was stronger than the promoting effect of replanted soil on FO proliferation. Furthermore, the number of PS in NP soil was significantly lower than that in TP soil at 0 DAP, while the number of FO in NP soil was significantly greater than that in 1/3TP, 2/3TP and TP soils at 0 DAP. The changes were obviously different from that at 9 DAP. These results are different from those in the field [10], which might be related to the difference between the air-dried soil used in this experiment and the field soil after fallow cultivation in the literature [51]. A valuable clue was thus presented that soil may have a strong ability restored microbe balance in rhizosphere soil during the fallow stage.

### 3.2. NB-LRRs Failed to Respond Timely and Effectively to Pathogenic Fusarium oxysporum in Replanted R. glutinosa

Pathogens almost always occupy extracellular niches [37]. Early studies had unraveled that *Fusarium oxysporum* could inhibit the release of ATP after invading the host plant [52], which is required for NB-LRR activation [37], but low doses toxins secreted by FO could induce the biosynthesis of phytoalexins in the host plants [53]. In this study, the upregulated expressions of *NB-LRRs* were mainly appeared at 3 DAP in TP and at 6 DAP in NP, which were consistent with the changes of the FO relative values (Figure 5). According to the original data associated with the relative values, we found that the upregulated expression of *NB-LRRs* was the strongest when the number of FO was at the lowest level (about 0.95 × 10^8^ cell·g^−1^ soil), and gradually decreased with the increasing of FO number in six days after transplanting. These results exhibited that *NB-LRR* expressions have a typical regulars with “promotion in low concentration and suppression in high concentration” when countered FO invasion, which was consistent with Scott et al. (2019) [54].

In antagonistic associations with microbes, plants have evolved to form two strategies of PTI and ETI for fighting microbial pathogens [37]. A well-known view is that immune receptor play a vital role in the recognition to pathogens [55,56], and constitutive downstream proteins are tightly controlled by both positive and negative regulators [39,57]. An integrated understanding for plant immune response is thereby in both the immune receptor and downstream signal transduction. Based on studies till date, there is mounting evidence that two possible interpretations were supported for the relationships of immune response and plant death. (i) Plant immunity has not been triggered [37,38]. (ii) Excessive immunity response often leads to inhibition of normal plant growth and even death [57,58,59]. In this study, the majority of *NB-LRRs* identified in replanted *R. glutinosa* roots were upregulated or downregulated in TP at 3 DAP in comparison to 0 DAP. At same time, there were significantly positive correlations found between 12 *NB-LRRs* and the FO numbers in NP, 1/3TP, 2/3TP and TP during 0~3 DAP, but the FO numbers in 1/3TP, 2/3TP and TP were no different with those in NP at 3 DAP even significantly lower at 0 DAP. In addition, accompanied by the death of replanted *R. glutinosa* at 6 DAP, only 1 *NB-LRR* was upregulated (downregulated at 3 DAP) and 29 *NB-LRRs* were downregulated. Taking together these data showed that the stage of 0~3 DAP was the key stage for *NB-LRRs* to respond to replant disease stress, while the immune response was obviously inactivated to FO rather than excessive responses of NB-LRRs. Therefore, it was one of important reasons that *NB-LRRs* were not consecutively responsive to the FO proliferation at the transcriptional level in replanted *R. glutinosa* roots. These new findings provide insights into the response mechanism of *R. glutinosa* to replant disease.

One of the big gaps in our understanding of plant immunity is in the downstream signaling pathways after receptor protein activation [37]. Only two identified downstream signaling proteins of EDS1 and non-race-specific disease resistance 1 (NDR1) are required for signaling of all TIR-NB-LRRs and some CC-NB-LRRs, respectively [37]. For NB-LRR, the protein structure consists of a carboxy-terminal LRR domain for effector recognition, NB-ARC (nucleotide-binding adaptor shared by APAF-1, R proteins, and CED-4) and amino-terminal Toll/interleukin-1 receptor (TIR) or Coiled-coil (CC) domains for signal transduction to downstream proteins [41]. In this study, the six downregulated *NB-LRRs* (*RgNB5*, *RgNB14*, *RgNB26*, *RgNB29*, *RgNB34* and *RgNB35*) in TP at 3 DAP, were screened as candidates for responding to *R. glutinosa* replant disease. According to their functional conservation in these NB-LRRs, there were seven resistance in linkage group 1A (R1A), 2 resistance in linkage group 1B (R1B), 1 resistance to *Pseudomonas syringae* (RPS2), 1 target of AvrB operation 1 (TAO1) and 2 resistance to *powdery mildew* 8 (RPW8). Based on studies till date, EDS1 were required for downstream signaling of these identified NB-LRRs except for RPS2 (RgNB32) [60,61,62,63]. The results provide valuable clues for studying the signaling pathways that operate downstream of NB-LRR protein activation in replanted *R. glutinosa*.

### 3.3. Lower Level of SA Biosynthesis Stimulated by Fusarium oxysporum Might Be Cloesly Related to the Formation of R. glutinosa Replant Disease

Generally, the stress factors associated with replant disease were complex and multiple, including pathogens, nematode and abiotic stress [7]. To resist these stress factors, plant hormones are widely involved in resistance levels as important signaling molecules [64]. ABA is mainly associated with abiotic environmental stresses [44]. SA is typically involved in the defense against biotrophs [65]. JA and ET are generally thought to act together, and to play core roles in the defense against necrotrophs [66]. The relationship between the SA and JA-ET pathway is more antagonistic other than cooperative [44]. In this study, the four hormones were significantly activated under replant disease stresses compared with the control NP at 3 DAP, and presented each other significant positive correlation (*p* < 0.001). The cooperation for SA and JA-ET was consistent with some literatures, such as Wu et al. (2018) [67] and Adie et al. (2007) [68], which was different from the antagonistic relationship between SA and JA-ET. A reasonable explanation for the cooperation of ABA with SA was that ABA increased susceptibility to pathogens in some plant-pathogen interactions [45]. However, these inferences need to be further verified. Previous studies displayed that some plant hormones including ET and ABA involved in the formation mechanism of *R. glutinosa* replant disease and closely related to immune resistance [48,69]. Some studies have demonstrated that SA was involved in the defense against *Fusarium oxysporum* and associated with acteoside accumulation, one of the important pharmacodynamic component [46,47]. More importantly, there were significant negative correlations only between ∆FO and ∆SA with the gradient changes in the stress level of replant disease (Table 4). The results revealed that the SA synthesis, which may be inhibited by *Fusarium oxysporum*, was involved in the formation of replant disease in *R. glutinosa*.

Based on the above results, a possible depiction of the immune response and its potential crosstalk with microbes and plant hormones in replanted *R. glutinosa* was drawn, and it is shown in Figure 6. These findings provide insights into the formation of replant disease.

## 4. Materials and Methods

### 4.1. Plant Growth and Treatments

To obtain aseptic plantlets, tuberous roots of *R. glutinosa* “Wen 85-5” were surface sterilized with 0.1% mercuric chloride solution for 17–20 min, washed five times with the sterile water, and then cultured in sterile bottles with two layers of damp gauze at the bottom. The shoots, approximately 1 cm long, were cut and cultured on hormone-free MS agar medium containing 30 g·L^−1^ sucrose and 10 g·L^−1^ agar [47]. The explants were cultured under controlled conditions (25 °C, 4000 lux, 14 h light/10 h dark photoperiod) in a growth chamber for 30 days.

To enhance the quality of plantlets for transplantation, the aseptic plantlets of *R. glutinosa* with seven to eight leaves were adapted in a phytotron (28 °C, 10,000 lux, 14 h light/10 h dark photoperiod) for 12 h, followed by unscrewing the bottle caps with a small opening to adapt for 6 h, and then removing the cap to adapt for 30 h (Figure 7A). After carefully washing away the adherent medium on the roots, the plantlets were adapted in sterile water for one day and transplanted into plastic pots.

Pot experiments were performed under controlled conditions (28 °C, 10,000 lux, 14 h light/10 h dark photoperiod) at the Institute of GAP for Chinese Medicinal Materials, Fujian Agriculture and Forestry University. *R. glutinosa* plantlets after acclimatization were transplanted on 23 August 2018 and grown in plastic pots of 18 cm diameter and 15 cm height (1.38 kg soil per pot). Three plants were planted as three replicate sub-samples in each pot. Four treatments of replant disease levels were constructed by mixing two kinds of soils in different proportions. The soils were collected from the site where *R. glutinosa* had not been planted for at least 10 years (NP) and where *R. glutinosa* had been consecutively planted for three years (TP) in Wen County, Jiaozuo City, Henan Province, in the “geo-authentic” zone of *R. glutinosa* cultivation (34°56′ N, 112°58′ E). The air-dried soil samples were taken to the laboratory for this experiment. Four treatments thus included NP, 1/3TP (mixed by 2 NP soils and 1 TP soil), 2/3TP (mixed by 1 NP soil and 2 TP soils), and TP, NP of which was used as the control (Figure 7B).

### 4.2. The Collection of Fresh Root and Rhizosphere Soil Samples

On day zero, three after acclimatization (DAA), the fresh roots were collected after carefully washing with sterile water and drying with absorbent paper. The samples of 0 DAP and 3 DAA are the same. At 3, 6 and 9 DAP, the fresh roots and their rhizosphere soil were carefully collected as described in Wu et al. (2015) [10]. Briefly, the roots and the soil around the roots were carefully dug up using a sterilized fork spade and slightly shaken to remove loosely attached soil. The rhizosphere soil that was tightly attached to roots (1–3 mm zone around the root) was brushed off and collected. All of the collected samples were immediately frozen in liquid nitrogen and then stored at −80 °C for further experiments for soil DNA extraction, absolute quantification of PS and FO, qRT-PCR of NB-LRR and measurement of plant hormones and physiological index.

### 4.3. The Extraction of Soil DNA and Its Method Comparison

Approximately 5 g of soil of each sample was weighed for the extraction of soil DNA. Three extraction methods were compared according to the electrophoretic strips. Method I referred to the conventional cetyltrimethylammonium bromide (CTAB) method [70]. Methods II and III used different extraction methods based on the optimization for removing humic acid. PCR and gel electrophoresis were used to evaluate the different DNA extraction methods for PS and FO. For PCR, specific primers of PS (PS for: 5′-GGTCTGAGAGGATGATCAGT-3′, PS rev: 5′-TTAGCTCCACCTCGCGGC-3′) and FO (ITS1-F: 5′-CTTGGTCATTTAGAGGAAGTAA-3′, AFP308R: 5′-CGAATTAACGCGAGTCCCAA-3′) were synthesized with reference to Wu et al. (2015) [10] (SunYa Biotechnology Co., Ltd. Fuzhou, China). Conventional PCR was performed using a Thermo cycler instrument (ThermoFisher Scientific A24812, Waltham, MA, USA) to detect transcript abundance. Each 20-μL reaction contained 0.4 μM each primer, 0.5 U 2× Es taq MasterMix enzyme, cDNA and nuclease-free water. The amplification procedure was 95 °C for 2 min, 95 °C for 5 s, and 58 °C to anneal for 30 s, for 30 cycles. Horizontal gel electrophoresis (1% gel) was used to evaluate the effect of the extraction with a DL2000 DNA Marker (Takara, Japan) in 130 V, 200 mA (Liuyi DYY-12, Beijing, China). The results showed that the extraction effect of method III was relatively best (Appendix A).

### 4.4. Absolute Quantification of Pseudomonas spp. and Fusarium oxysporum

To accurately evaluate the shift of rhizosphere microorganisms to avoid the expression changes of reference genes in different growth phases and soil samplings [71], the characteristic rhizosphere microorganisms of beneficial PS and pathogenic FO were detected based on absolute quantification PCR (AQ-PCR). 

#### 4.4.1. Construction of the Recombinant Plasmid

Amplifications of PS and FO specific primers were performed by conventional PCR (as mentioned above) and touchdown PCR (the same 20-μL reaction; 95 °C for 2 min, 95 °C for 5 s, 50–60 °C to anneal for 30 s, 72 °C for 30 s, for eight cycles; 95 °C for 2 min, 95 °C for 5 s, 58 °C to anneal for 30 s, for 30 cycles), respectively (Appendix A). The bright electrophoretic strips were cut and extracted using gel pure DNA kits following the manufacturer’s instructions (Magen D2111-02, China). The gel extraction solutions were concentrated to over 50 ng·μL^−1^ of cDNA using a concentrator for approximately 25 min at 1400 rpm (Eppendorf Concentrator Plus AG5305, V-AQ mode). DNA fragments were inserted into a vector and ligated overnight at 16 °C using the pMD19-T vector cloning kit following the manufacturer’s instructions (Takara 6013, Japan). Then, 5 μL of vector DNA solution and 50 μL of *E. coli* DH 5α were blended and incubated in ice for 30 min. After heat shock at 42 °C for 60 s, the vectors were kept on ice for 3 min. The solution with 600 μL of liquid LB culture medium was closed using Parafilm and the culture was shaken for 60 min (37 °C, 200 rpm). Then, 45 μL of 5-Bromo-4-chloro-3-indolyl β-d-galactoside (X-Gal), 10 μL of isopropyl-l-d-thiogalactopyranoside (IPTG) and 200 μL of the culture solution were smeared evenly on solid LB culture medium containing 0.4% Ampicillin (Amp) and then incubated at 37 °C for 14 h. We selected 1–2 white single colonies and added 300 μL of liquid LB culture medium containing 0.4% Amp for co-culture (37 °C, 200 rpm) for approximately 14 h until the solutions were turbid. PCR and 1% gel electrophoresis as aforementioned were used to identify the size of DNA fragment comparing with a DL2000 DNA Marker (Takara, Japan). The solutions containing the appropriate size of DNA fragments were chosen to extract plasmids using Hipure Plasmid Micro Kit following the manufacturer’s instructions (Magen, China). 10 μL solutions of plasmids were used for sequencing in Biosune, China (987 bp for PS, 433 bp for FO; Appendix A).

#### 4.4.2. Establishment of Standard Curve

The plasmid solutions of s containing the right size of DNA fragments were amplified again using plasmids primer RV-M/M13-47 by qRT-PCR (BIO-RAD CFX96, USA) (20-μL reaction contained 0.4 μM each primer, 0.5 U SYBR Premix EX Taq II (2×), cDNA and nuclease-free water following the Takara RR820A instructions; 95 °C for 2 min, 95 °C for 5 s, 55 °C to anneal for 30 s, 72 °C for 30 s, for 30 cycles). The DNA concentration of target genes were detected by a NanoDrop2000 spectrophotometer (Thermo Scientifi, USA) and then diluted to 0, 1, 2, 3, 4 ng/μL. The standard curves were drawn based on the DNA concentration of target genes and Ct values (Appendix A). The calculation of plasmid copy number was based on Shirima et al. (2017) [72].
(1)$$\mathrm{Plasmid}\;\mathrm{copy}\;\mathrm{number}\;=\frac{6.02\times 10^{23}\times \left(\mathrm{copies}\cdot {\mathrm{mol}}^{-1}\right)\times \mathrm{plasmid}\;\mathrm{amount}\left(g\right)*}{\mathrm{MW}}$$
MW = plasmid molecular weight, (=plasmid size (2692 bp) × molar mass per base (660 g·mol^−1^·bp^−1^); 6.02 × 10^23^ molecules/mole = Avogadro’s constant; * Plasmid amount was calculated from the plasmid concentration determined by a NanoDrop2000 spectrophotometer (Thermo Scientifi, USA).

#### 4.4.3. Determination of AQ-PCR

Ct values of recombinant plasmid containing 5 gradient concentration (as control) and soil DNA extracts were detected using specific primers of PS and FO by qRT-PCR (20-μL reaction was as mentioned above). The copy numbers of PS and FO were calculated based on the standard curve. All reactions were replicated three times.

### 4.5. qRT-PCR Analysis of NB-LRRs

For 35 previously identified *NB-LRRs* (Appendix A), RNA extraction of roots, reverse transcription and qRT-PCR analysis (BIO-RAD CFX96, USA) were conducted as described by Chen et al. (2018) [41]. All reactions were replicated three times. The data were normalized on the basis of the 18S rRNA threshold cycle (Ct) value. The samples with the NP treatments were used as the controls at the same sampling time, and their normalized Ct values were set to 1. The relative gene expression of the other treatments was calculated using the 2^−∆∆CT^ method [73].

### 4.6. Measurement of ABA, SA, ET and JA

The contents of abscisic acid (ABA), salicylic acid (SA), ethylene (ET) and jasmonate (JA) were determined using a one-step double-antibody sandwich enzyme-linked immunosorbent assay (ELISA). Briefly, 1.0 g of fresh root was ground in 5 mL of phosphate-buffered saline (PBS) (0.01 M, pH 7.4) with an ice-cooled mortar and centrifuged at 2500 rpm for 20 min at 4 °C to obtain a supernatant for the ELISA analysis following the protocol described in Zhao et al. (2006) [74]. The mouse monoclonal antigen and antibodies against free ABA, SA, ET and JA were provided by MLBIO Co. Ltd., Shanghai, China. The hormone content was measured at 450 nm using a microplate reader (BIO-Tek ELX800, USA). Calculations of the ELISA data were performed as described in Wang et al. (2012) [75]. The recovery percentages obtained by using internal standards during extraction and analysis were all >90%.

### 4.7. Measurement of Root Activity and the Physiological Index

To determine root activity, the methodology described by Zhang et al. (2013) [76] was followed with modifications. Approximately 0.5 g of fresh root was mixed with 10 mL of a half-and-half blend of 0.4% triphenyl tetrazolium chloride (TTC) and 1/15 M PBS (pH 7.4) and incubated at 37 °C for 1 h, and then the reaction was stopped by 2 mL of 1 M H_2_SO_4_. The roots were homogenized in ethyl acetate with a capacity of 10 mL. The absorbance of the final solution was measured at 415 nm (Pgeneral T6-1650E, China). A standard curve was used to determine the concentration of root activity in the extract. 

The determination of SOD, POD and CAT activities and MDA content were as described by Li et al. (2017) [48] and Deenamo et al. (2018) [77], respectively. The pretreatment was the same, and then 0.5 g of roots was homogenized in 5 mL of precooled PBS (0.05 M, pH 7.8) with a small amount of quartz sand. Extracts were centrifuged for 15 min at 13 000 rpm. The supernatant was used for the measurement of the four indexes. The colorimetric wavelengths were 560 nm, 470 nm and 240 nm for SOD, POD and CAT, and 600 nm, 532 nm, 450 nm for MDA (Pgeneral T6-1650E, China).

The H_2_O_2_ content was determined by the KI method [78]. In short, 0.2 g of roots was homogenized in 0.8 mL of precooled 0.1% TCA (trichloroacetic acid) with liquid nitrogen. Extracts were centrifuged for 20 min at 19,000 rpm. Then, 0.5 mL of supernatant was added to 2 mL of KI (1 M) and 0.5 mL of PBS (100 M) for reaction at darkness for 1 h. The absorbance of the final solution was measured at 390 nm (Pgeneral T6-1650E, China). A standard curve was used to determine the concentration of H_2_O_2_ in the extract.

### 4.8. Statistical Analysis

Multiple comparisons (LSD) and Pearson correlation were analysed with SAS statistical software (V9.1 SAS Institute Inc., Cary, NC, USA). Each value with three replicates represented as the mean ± SD. *p* < 0.05 was considered as significant between any two groups. A heatmap was generated via the hierarchical clustering method using the MeV 4.9.0 tool [79].

## 5. Conclusions

Our results indicated that *R. glutinosa* replant disease promoted *Fusarium oxysporum* proliferation in rhizospheres soil, but *NB-LRRs* were not consecutively responsive to the FO proliferation at the transcriptional level in *R. glutinosa* roots. The analysis on the relationships between *Fusarium oxysporum* numbers and *NB-LRRs* expression showed that the *NB-LRRs* responded to the *Fusarium oxysporum* invasion with a typical phenomenon of “promotion in low concentration and suppression in high concentration”, and 6 *NB-LRRs* were identified as candidates for responding to *R. glutinosa* replant disease. Salicylic acid (SA), jasmonic acid (JA), ethylene (ET) and abscisic acid (ABA) involved in the formation mechanism of *R. glutinosa* replant disease. Importantly, salicylic acid, as an important signaling molecule, which may be inhibited by *Fusarium oxysporum*, was involved in the formation of replant disease in *R. glutinosa*. Concomitantly, the physiological indexes acting downstream of life activities were affected by complex replant disease stresses and exhibited strong fluctuations, resulting in the death of *R. glutinosa*. These findings provide important insights and clues into further revealing the mechanism of replant disease.

## Figures and Tables

**Figure 1 ijms-20-03203-f001:**
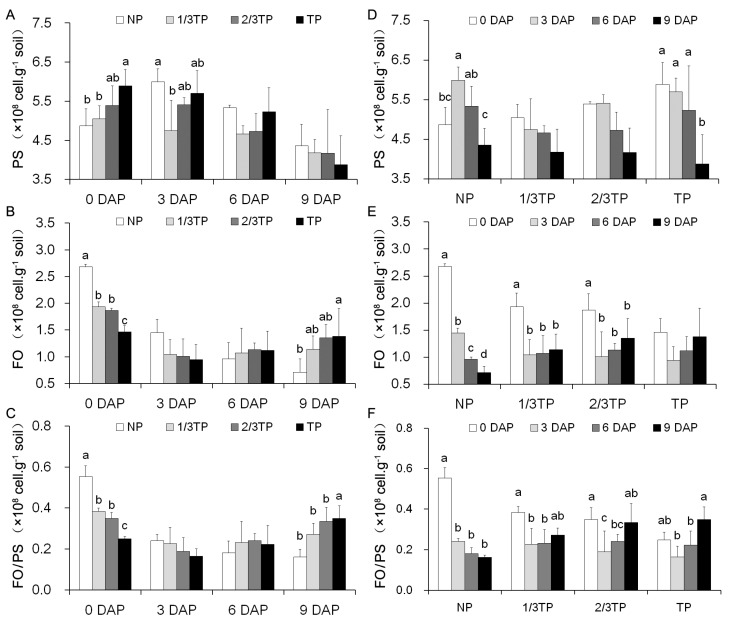
The influences of different replant disease stress levels on the numbers and ratio of FO to PS in *R. glutinosa* rhizosphere soils over time. NP, 1/3TP, 2/3TP and TP are the gradient treatments of replant disease stresses. Four samples in each group are compared with NP (**A**–**C**) and 0 DAP (**D**–**F**) respectively, and different lower-case letters indicate significant differences (*p* < 0.05; LSD). Data represented as the mean ± SD (*n* = 3). DAP: Days after planting. PS: *Pseudomonas* spp. FO: *Fusarium oxysporum*. NP: Soil that was never planted with *R. glutinosa* for at least 10 years. TP: Soil that was consecutively planted with *R. glutinosa* in the same soils for three years.

**Figure 2 ijms-20-03203-f002:**
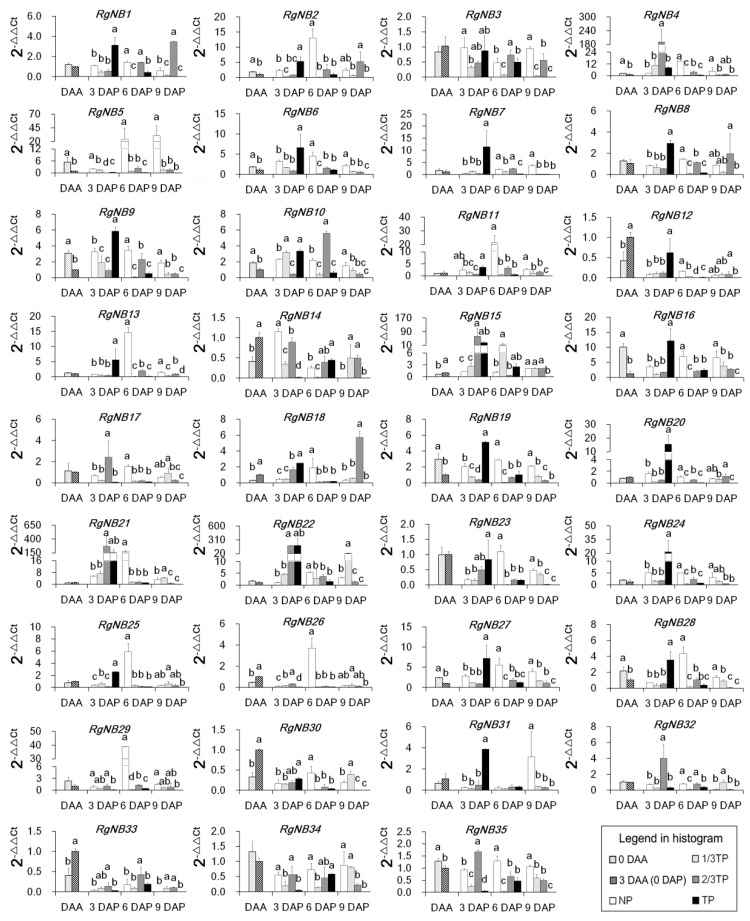
Nucleotide binding-leucine-rich repeats (*NB-LRRs*) expression in *R. glutinosa* roots during acclimatization and planting phase with different replant disease stress levels. The ratios of 0 DAA, 3 DAA (0 DAP), NP, 1/3TP, 2/3TP and TP gradient treatments to 3 DAA (0 DAP) are calculated and shown. Two or four samples in each group are compared with 3 DAA and NP respectively, and different lower-case letters indicate significant differences (*p* < 0.05; LSD). Data represented as the mean ± SD (*n* = 3). DAA: Days after acclimatization. DAP: Days after planting. NP: Soil that was never planted with *R. glutinosa* for at least 10 years. TP: Soil that was consecutively planted with *R. glutinosa* in the same soils for three years.

**Figure 3 ijms-20-03203-f003:**
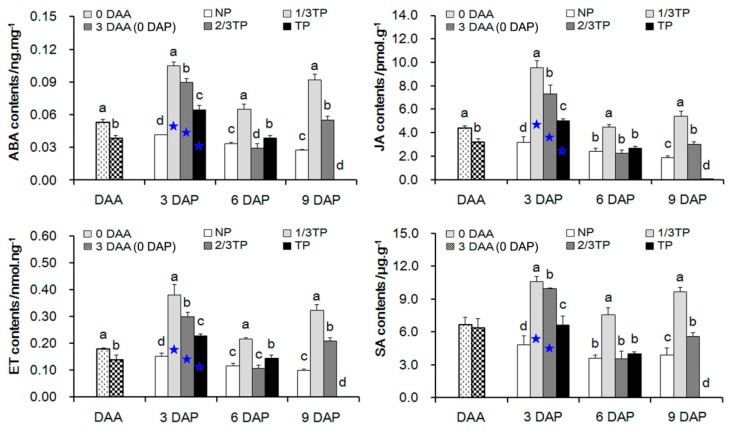
The four plant hormone contents in *R. glutinosa* roots during acclimatization stage and planting stage with different replant disease stress levels. 0 DAA, 3 DAA (0 DAP), NP, 1/3TP, 2/3TP and TP are gradient treatments. Two or four samples in each group are compared with 3 DAA (0 DAP) and NP respectively, and different lower-case letters indicate significant differences (*p* < 0.05; LSD). 3 DAP with NP, 1/3TP, 2/3TP and TP gradient treatments is compared with 3 DAA (0 DAP), and the asterisk indicate significant differences (*p* < 0.05; LSD). Data represented as the mean ± SD (*n* = 3). DAA: Days after acclimatization. DAP: Days after planting. NP: Soil that was never planted with *R. glutinosa* for at least 10 years. TP: Soil that was consecutively planted with *R. glutinosa* in the same soils for three years.

**Figure 4 ijms-20-03203-f004:**
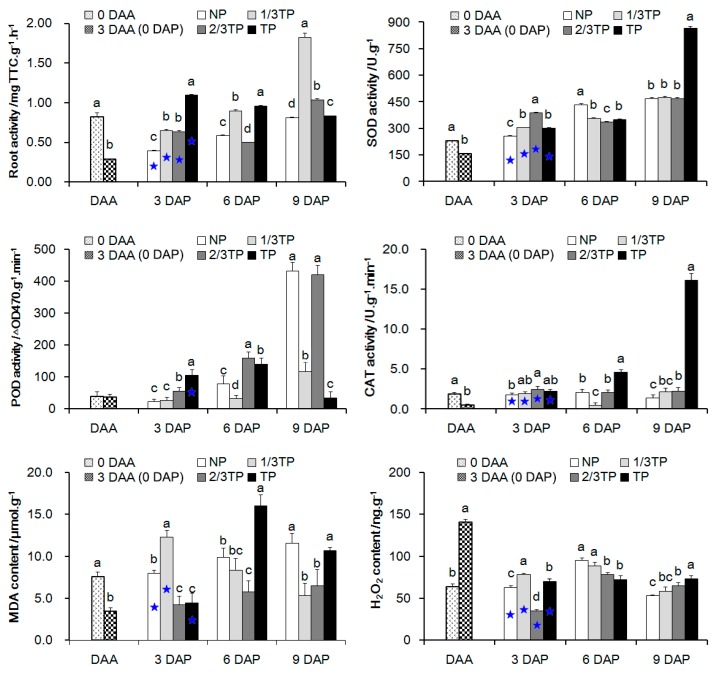
The contents of physiological indexes in *R. glutinosa* roots during acclimatization stage and planting stage with different replant disease stress levels. 0 DAA, 3 DAA (0 DAP), NP, 1/3TP, 2/3TP and TP are gradient treatments. Two or four samples in each group are compared with 3 DAA (0 DAP) and NP respectively, and different lower-case letters indicate significant differences (*p* < 0.05; LSD). 3 DAP with NP, 1/3TP, 2/3TP and TP gradient treatments is compared with 3 DAA (0 DAP), and the asterisk indicate significant differences (*p* < 0.05; LSD). Data represented as the mean ± SD (*n* = 3). DAA: Days after acclimatization. DAP: Days after planting. NP: Soil that was never planted with *R. glutinosa* for at least 10 years. TP: Soil that was consecutively planted with *R. glutinosa* in the same soils for three years.

**Figure 5 ijms-20-03203-f005:**
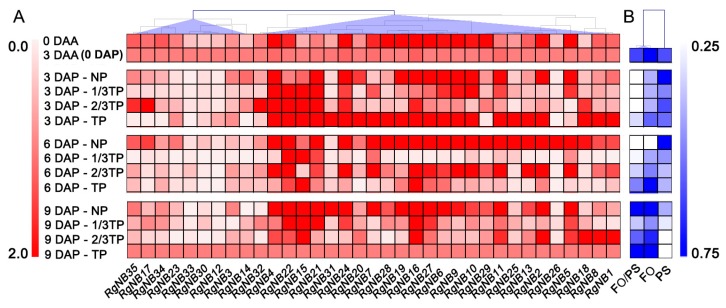
The heatmaps of the 35 *NB-LRR* expressions in *R. glutinosa* roots and the numbers of FO and PS and their ratios in *R. glutinosa* rhizosphere soils. (**A**) The expression profiles of 35 *NB-LRRs* in *R. glutinosa* roots over time. 3 DAA (0 DAP) is the control; (**B**) the changes in relative values of FO, PS and FO/PS over time. Calculation of the relative values: The cell numbers and the FO/PS values are first compared with that at 0 DAP (3 DAA), and then normalized (*Z*-score) for each microorganism respectively to display in the same color scale. DAA: Days after acclimatization. DAP: Days after planting. NP: Soil that was never planted with *R. glutinosa* for at least 10 years. TP: Soil that was consecutively planted with *R. glutinosa* in the same soils for three years.

**Figure 6 ijms-20-03203-f006:**
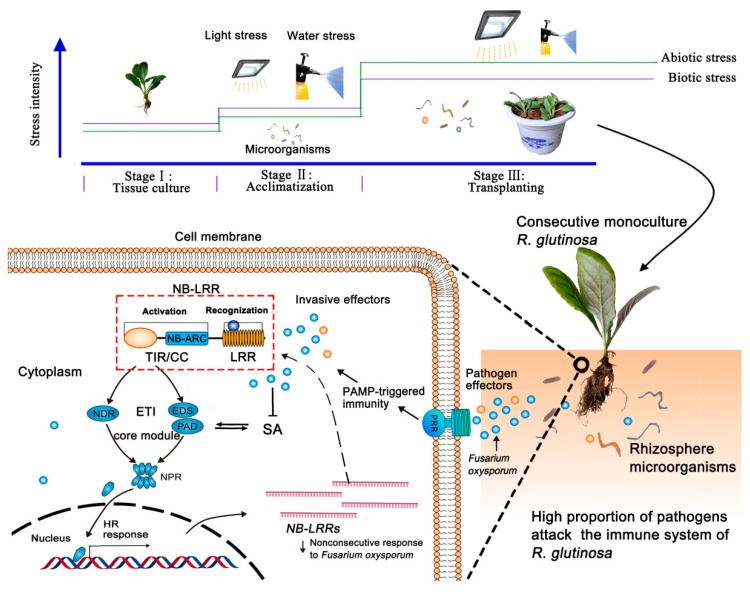
Schematic of effector-triggered immunity response mediated by consecutive monoculture stress in *R. glutinosa*.

**Figure 7 ijms-20-03203-f007:**
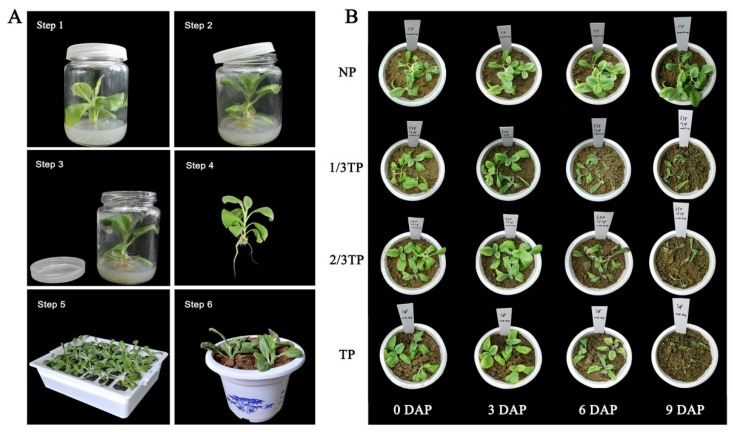
The operation process for acclimatization and the phenotype changes of *R. glutinosa* in the experiment. Obvious irreversible injury occurred at 6 DAP until death at 9 DAP. (**A**) The six main steps of acclimatization and transplanting. Step 1: Closed adaptation for 12 h; Step 2: Small opening for 6 h; Step 3: Completely open for 30 h; Step 4: Washing the culture medium carefully with sterile pure water; Step 5: Plantlets were adjusted to sterile pure water for 24 h; Step 6: Transplanting three plants per pot; (**B**) the phenotype changes of *R. glutinosa* under different replant disease stresses within 9 DAP. In the pot stage, 1/3TP, 2/3TP and TP are compared with NP, and 3 DAP, 6 DAP and 9 DAP are compared with 0 DAP (3 DAA). DAP: Days after planting. NP: Soil that was never planted with *R. glutinosa* for at least 10 years. TP: Soil that was consecutively planted with *R. glutinosa* in the same soils for three years.

**Table 1 ijms-20-03203-t001:** Pearson correlations among the different replant disease stress levels and the changes in amplitude of FO/PS and the numbers of PS and FO within 3 DAP.

	∆FO/PS	∆PS	∆FO
Levels of replant disease stresses	0.7918 **	−0.5611	0.6809 *

* *p* < 0.05, ** *p* < 0.01; ∆ represents the change from 0 to 3 DAP.

**Table 2 ijms-20-03203-t002:** Pearson correlations among the FO/PS and the numbers of PS and FO within 9 DAP.

	PS	FO
FO/PS	−0.2488	0.9208 **
PS		0.1331

** *p* < 0.01.

**Table 3 ijms-20-03203-t003:** Pearson correlations among the expression of 35 NB-LRRs and the variation of PS and FO numbers from 0 DAP to 3 DAP.

	*RgNB1*	*RgNB2*	*RgNB3*	*RgNB4*	*RgNB5*	*RgNB6*	*RgNB7*
PS	−0.1156	0.0187	0.6833	−0.1978	0.5612	−0.0038	−0.3908
FO	0.9737	0.9344	0.4558	−0.3259	−0.4752	0.9219	0.9963 **
	*RgNB8*	*RgNB9*	*RgNB10*	*RgNB11*	*RgNB12*	*RgNB13*	*RgNB14*
PS	−0.2584	0.0342	−0.1756	0.1503	−0.4000	−0.2856	0.7931
FO	0.9950 **	0.8935	0.5160	0.8193	0.9973 **	0.9984 **	−0.7543
	*RgNB15*	*RgNB16*	*RgNB17*	*RgNB18*	*RgNB19*	*RgNB20*	*RgNB21*
PS	−0.3292	−0.1421	0.0665	−0.4500	−0.0412	−0.2643	−0.3687
FO	0.0526	0.9794 *	−0.4775	0.8240	0.9452	0.9970 **	0.2326
	*RgNB22*	*RgNB23*	*RgNB24*	*RgNB25*	*RgNB26*	*RgNB27*	*RgNB28*
PS	−0.4505	−0.4356	−0.1776	−0.3769	−0.1168	−0.0632	−0.2509
FO	0.5924	0.8712	0.9858 *	0.9845 *	−0.6370	0.9564 *	0.9953 **
	*RgNB29*	*RgNB30*	*RgNB31*	*RgNB32*	*RgNB33*	*RgNB34*	*RgNB35*
PS	0.4527	−0.3214	−0.3369	−0.0666	−0.4524	0.6505	0.3434
FO	−0.8157	0.9787 *	0.9965 **	−0.3699	−0.5155	−0.7435	−0.6044

* *p* < 0.05; ** *p* < 0.01.

**Table 4 ijms-20-03203-t004:** Pearson correlations among the variation of the PS, FO, jasmonic acid (JA), abscisic acid (ABA), ethylene (ET) and salicylic acid (SA) contents under replant disease stresses from 0 to 3 DAP.

	∆FO	∆JA	∆ABA	∆ET	∆SA
∆PS	−0.7327	0.1328	0.0254	0.1123	0.2366
∆FO		−0.6524	−0.5981	−0.6560	−0.7475 *
∆JA			0.9752 **	0.9839 **	0.9182 **
∆ABA				0.9650 **	0.9543 **
∆ET					0.9009 **

* *p* < 0.05, ** *p* < 0.001; ∆ represents the variation in content.

**Table 5 ijms-20-03203-t005:** Pearson correlations among the variation of the two microorganisms and the six physiological indexes under replant disease stresses during three and six days after transplanting.

		∆Root Activity	∆SOD	∆POD	∆CAT	∆H_2_O_2_	∆MDA
0~3 DAP	∆PS	0.0357	0.2189	0.3088	−0.1421	−0.3443	−0.3313
	∆FO	0.4389	−0.1874	0.2671	0.4416	0.1959	−0.1724
0~6 DAP	∆PS	0.1759	−0.0713	−0.0387	−0.2458	0.1394	−0.2020
	∆FO	0.2250	0.1574	0.2263	0.7464 *	−0.5732	0.7316 *

* *p* < 0.05; ∆ represents the variation in content.

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
