# Peer review of "NB-LRRs Not Responding Consecutively to Fusarium oxysporum Proliferation Caused Replant Disease Formation of Rehmannia glutinosa"

_ijms, 2019, doi:10.3390/ijms20133203_

Reviewer 1 Report

Microbes either pathogen or beneficial ones significantly influences plants growth while underlying mechanism remains unclear.  Therefore, it merits research to investigate different ways of their interactions including model plant pathogens using molecular approaches. Overall, we know much less about the real world solution of plant pathogen problem to enhance plant performance. This paper has studied the role of NB-LRRs in determining plant responses to Fusarium. That is a well-performed study, and the manuscript provides new information to the existing knowledge of plant-pathogen interactions as mediated by the NB-LRRs. The manuscript is clear in its contents and may be considered for publication after substantial revision. Below are my suggestions and recommendations that may enhance the readership of this work.

 Ln 16. “Leads to the emergence of replant disease”

Ln 46. “shape rhizosphere microbial communities” to “shape rhizosphere microbial community composition and diversity ([14-18…. Theories, mechanisms and patterns of microbiome species coexistence in an era of climate change. In Microbiome community ecology (pp. 13-53). Springer, Cham.).  

Ln 52. “dverse rhizosphere microbes’ confusing phrase, consider to revise it or make is clear

Ln 57. “However, little is known about the variation of FO numbers in R. glutinosa rhizosphere soil under 57 replant disease stress.” I saw some recent papers describing the FO under same conditions and author statement may be over-stating. E.g. Applied soil ecology, 130, 271-279 (2018); Scientific reports 5 (2015): 15871.

Ln 60. “composition and diversity of defence genes” I think it should me “microbiome” or “microbes” instead of “genes”.

Ln 70. “fast-evolving effectors [37 with….] or with [55…] at ln 284.” authors may need to support their views with recent literature (European journal of plant pathology, 149(3), 779-786 (2017).

Ln 107. “changes in the characteristic microorganisms” confusing statement

Ln 126. Authors need to define “microbe balance”. What does it mean in this study?

Ln 188.  “Notably, there were significant negative correlations between FO and SA (P<0.05) and significant positive correlations among the four hormones (P<0.01) (Table 4)” very interesting results, authors need to elaborate these with reason in the abstract to increase the readership of this work.

Ln 259. “Microbial plant pathogens” I guess all pathogens are microbial?

 Ln. “both positive regulators and negative regulators” to “both positive and negative regulators”

  Author Response

Dear Prof:

 Thank you very much for your comments and suggestions for our works. We are very gratified that you had the patience and kindness to bear with our manuscript. According to your comments, all section of manuscript had been carefully revised. Revised parts have marked using the blue font in the revised manuscript. The point-by-point responses to your comments are as follows:

 1. Ln 16. “Leads to the emergence of replant disease”

Answer: Thanks for your suggestions. We have replaced "leads to replant disease" with “leads to the emergence of replant disease” in line 16 of page 1.

2. Ln 46. “shape rhizosphere microbial communities” to “shape rhizosphere microbial community composition and diversity ([14-18…. Theories, mechanisms and patterns of microbiome species coexistence in an era of climate change. In Microbiome community ecology (pp. 13-53). Springer, Cham.).

Answer: Thanks for your suggestions. We have replaced "shape rhizosphere microbial communities" with “shape the composition and diversity of microbial community” and added the reference [19] in line 52 of page 2, and line 576~578 of page 19.

3. Ln 52. “adverse rhizosphere microbes’ confusing phrase, consider to revise it or make is clear

Answer: We have revised the confusing phrase to “the rhizosphere microecology catastrophe” in line 58 of page 2.

4. Ln 57. “However, little is known about the variation of FO numbers in R. glutinosa rhizosphere soil under replant disease stress.” I saw some recent papers describing the FO under same conditions and author statement may be over-stating.

Answer: Thanks for your suggestions. We have corrected the sentence to “However, it remains largely unknown that the mechanism how replanted R. glutinosa responded to the changes in composition and diversity of microbial community in rhizosphere soil” in line 62~64 of page 2.

5. Ln 60. “composition and diversity of defence genes” I think it should me “microbiome” or “microbes” instead of “genes”.

Answer: Thanks for your suggestions. We have replaced “defence genes” using “microbiome” in line 66 of page2.

6. Ln 70. “fast-evolving effectors [37 with….] or with [55…] at ln 284.” authors may need to support their views with recent literature (European journal of plant pathology, 149(3), 779-786 (2017).

Answer: We have added the reference [39] in line 76 of page 2, and line 295 of page 11.

7. Ln 107. “changes in the characteristic microorganisms” confusing statement

Answer: We have revised the sentence to “To understand the changes in the beneficial PS and pathogenic FO in rhizospheres of replanted R. glutinosa, the numbers of them were showed in Figure 1”. The revised contents were showed in line 113-114 of page 3.

8. Ln 126. Authors need to define “microbe balance”. What does it mean in this study?

Answer: We have modified using the accurate phrase. The revised content was as follows “restore balances among microbial communities” in line 131 of page 3.

9. Ln 188. “Notably, there were significant negative correlations between FO and SA (P<0.05) and significant positive correlations among the four hormones (P<0.01) (Table 4)” very interesting results, authors need to elaborate these with reason in the abstract to increase the readership of this work.

.Answer: Thanks for your suggestion. The related description have been modified to “Furthermore, four critical hormones of salicylic acid (SA), jasmonic acid (JA), ethylene (ET) and abscisic acid (ABA) had higher levels in 1/3TP, 2/3TP and TP than those in NP. And increasing extents of SA contents have significantly negative trends with FO changes, which implied that SA might be inhibited by FO in replanted R. glutinosain line 28-32 of abstract , and “Salicylic acid (SA), jasmonic acid (JA), ethylene (ET) and abscisic acid (ABA) involved in the formation mechanism of R. glutinosa replant disease. Importantly, salicylic acid, as an important signaling molecule, which may be inhibited by Fusarium oxysporum, was involved in the formation of replant disease in R. glutinosain line 516-519 of conclusions.

10. Ln 259. “Microbial plant pathogens” I guess all pathogens are microbial?

Answer: We have deleted “Microbial plant” in line 269 of page 10.

11. Ln289. “both positive regulators and negative regulators” to “both positive and negative regulators”

Answer: We have replaced “both positive regulators and negative regulators” with “both positive and negative regulators” in line 295 of page 11.

Reviewer 2 Report

The interaction between plants and their surrounding microbes is key to plant survival in a natural environment and the diverse mechanisms are very interesting in plant biology.  The manuscript by Chen et al. analyzed the responses of Rehmannia glutinosa to rhizosphere microbes, mainly in the context of NB-LRR gene expression, plant hormone content and the activity of physiologically related enzymes. Their work gives some new information of plant responses to the pathogenic rhizosphere strain Fusarium oxysporum. The general idea and organization of this work seems clear.

However, I personally do not agree with the conclusion “FO was not successfully recognized by the receptor NB-LRRs in R. glutinosa roots.”. To me, this is conceptually wrong. Basically, the authors performed qRT-PCR analysis to reveal the gene expression of NB-LRR genes during the acclimation process. That does not tell whether specific NB-LRRs “recognize” the pathogen at all. Their results could only give information on the up- and down of gene expression; that is to say whether the NB-LRRs genes were responsive to the pathogen in short or long-term stress at the transcriptional level. I think the authors should really pay attention to their interpretation and change their statement. For example, they can tune down their conclusion to “No NB-LRR genes were consecutively responsive to the pathogen proliferation at the transcriptional level.”. Accordingly, their schematic model in Fig.6 should be revised.  

Also, there are many grammatical mistakes and inappropriate descriptions throughout the manuscript. I strongly suggest the revise their text and language authors intensively and extensively, ideally with the help of a native English-speaker.  

Author Response

Dear Prof:

 Thank you very much for your comments and suggestions for our works. We are very gratified that you had the patience and kindness to bear with our manuscript. According to your comments, all section of manuscript had been carefully revised. Revised parts have marked using the blue font in the revised manuscript. The point-by-point responses to your comments are as follows:

 1. The interaction between plants and their surrounding microbes is key to plant survival in a natural environment and the diverse mechanisms are very interesting in plant biology.  The manuscript by Chen et al. analyzed the responses of Rehmannia glutinosa to rhizosphere microbes, mainly in the context of NB-LRR gene expression, plant hormone content and the activity of physiologically related enzymes. Their work gives some new information of plant responses to the pathogenic rhizosphere strain Fusarium oxysporum. The general idea and organization of this work seems clear.

I personally do not agree with the conclusion “FO was not successfully recognized by the receptor NB-LRRs in R. glutinosa roots.”. To me, this is conceptually wrong. Basically, the authors performed qRT-PCR analysis to reveal the gene expression of NB-LRR genes during the acclimation process. That does not tell whether specific NB-LRRs “recognize” the pathogen at all. Their results could only give information on the up- and down of gene expression; that is to say whether the NB-LRRs genes were responsive to the pathogen in short or long-term stress at the transcriptional level. I think the authors should really pay attention to their interpretation and change their statement. For example, they can tune down their conclusion to “No NB-LRR genes were consecutively responsive to the pathogen proliferation at the transcriptional level.”. Accordingly, their schematic model in Fig.6 should be revised.

Also, there are many grammatical mistakes and inappropriate descriptions throughout the manuscript. I strongly suggest the revise their text and language authors intensively and extensively, ideally with the help of a native English-speaker.

Answer: Thanks for your suggestion. We have corrected the inappropriate conclusion and revised the relative statement. The title was revised to “NB-LRRs not Responding Consecutively to Fusarium oxysporum Proliferation Caused Replant Disease Formation of Rehmannia glutinosa”.

“but FO was not successfully recognized by the receptor NB-LRRs in R. glutinosa roots” was modified to “NB-LRRs were not consecutively responsive to the FO proliferation at transcriptional levels” in line 24-25 of abstract and “but no NB-LRRs were consecutively responsive to the FO proliferation at the transcriptional level” in line 511-512 of conclusions.

“unrecognized the pathogenic FO” was revised to “might not responded to the pathogenic FO” in line 161-162 of page 5.

“3.2 NB-LRRs Fail to Recognize Pathogenic Fusarium oxysporum” was replaced with “3.2 NB-LRRs Failed to Respond Timely and Effectively to Pathogenic Fusarium oxysporum” in line 267 of page 10.

“Therefore, the failure of immune defence may be attributed mainly to the invalid recognition of NB-LRRs to pathogens in replanted R. glutinosa roots” was revised to “Therefore, it was one of important reasons that NB-LRRs were not consecutively responsive to the FO proliferation at the transcriptional level in replanted R. glutinosa roots” in line 309-310 of page 11.

The schematic model in Figure 6 has been revised in line 352 of page 13.

In addition, the language of this revised manuscript has been carefully edited.

Reviewer 3 Report

In the Abstract there is lack of the main methods that Authors applied. Please pay an attention that Abstract should contain: aim, main methodology and the most important conclusions.

Line 110: it should be: “ranged between 0 DAP...”

Line 120: put “evidenced” instead of “showed”

According IJMS rules the word: Figure and number should be bolded (in Fig. description) – please correct it in the whole ms text – the same “Table and number”

Figure 2 is totally illegible and should be improved.

Line 247: put: “revealed” instead of “showed”

Line 311: Pseudomonas syringae – should be written in capital letter

Lines 548, 656, 673, 679, 690, 693, 695, 699, 702 – rather old citations, I suggest to remove them or to replace by most newest one

Author Response

Dear Prof:

 Thank you very much for your comments and suggestions for our works. We are very gratified that you had the patience and kindness to bear with our manuscript. According to your comments, all section of manuscript had been carefully revised. Revised parts have marked using the blue font in the revised manuscript. The point-by-point responses to your comments are as follows:

 1. In the Abstract there is lack of the main methods that Authors applied. Please pay an attention that Abstract should contain: aim, main methodology and the most important conclusions.

Answer: Thanks for your suggestion. We have added “In this study, a controlled experiment was performed using different kinds of soils from sites never planted R. glutinosa (NP), replanted R. glutinosa (TP) and mixed by different ration of TP soils (1/3TP and 2/3TP), respectively.” in line 20-22 of abstract.

2. Line 110: it should be: “ranged between 0 DAP...”

Answer: Thanks for your suggestion. We have replaced “between 0 DAP and 9 DAP” with “ranged between 0 DAP and 9 DAP” in line 116 of page 3.

3. Line 120: put “evidenced” instead of “showed”

Answer: We have replaced “showed” with “evidenced” in line 126 of page 3.

4. According IJMS rules the word: Figure and number should be bolded (in Fig. description) – please correct it in the whole ms text – the same “Table and number”

Answer: Thanks for your suggestion. We have revised the format according IJMS rules in this paper.

5. Figure 2 is totally illegible and should be improved.

Answer: We have improved Figure 2 quality in line 170 of page 6.

6. Line 247: put: “revealed” instead of “showed”

Answer: We have replaced “showed” with “revealed” in line 257 of page 10.

7. Line 311: Pseudomonas syringae – should be written in capital letter

Answer: Thanks for your suggestion. We have replaced “pseudomonas syringae” with “Pseudomonas syringae” in line 322 of page 12.

8. Lines 548, 656, 673, 679, 690, 693, 695, 699, 702 – rather old citations, I suggest to remove them or to replace by most newest one

Answer: Thanks for your suggestion. We have used replaced the old citations with newest ones in line 562, 675, 692, 698, 701, 714, 717 and 720 of references, and revised the relative citation in line 338 of page 12 and line 482,485,493 of page 16.

 All of the revised contents were marked in blue colour in the text.

 We look forward to your earlier responses.

Yours sincerely,

Aiguo Chen, Mingjie Li and Zhongyi Zhang

E-mail: [email protected], [email protected], [email protected]

June 20, 2019

Round  2

Reviewer 2 Report

The authors addressed all my points in an appropriate way. So the manuscript may be OK for further production.